# A Near-Surface Layer Heat Treatment of Die Casting Dies by Means of Electron-Beam Technology

Torsten Schuchardt [1] , Sebastian Müller [2,*] and Klaus Dilger [1,*]

1 Institute of Joining and Welding, Technische Universität Braunschweig, 38106 Braunschweig, Germany; t.schuchardt@tu-braunschweig.de
2 Institute of Casting Technology, Friedrich-Alexander-Universität Erlangen-Nürnberg, 90762 Fürth, Germany
\* Correspondence: seb.mueller@fau.de (S.M.); k.dilger@tu-braunschweig.de (K.D.)

**Abstract:** Increasing the service life of die casting dies is an important goal of the foundry industry. Approaches are either material- or process-related. Despite new material concepts, hot work steels such as H11 are still predominantly used in the uncoated condition for die casting dies. In order to withstand the stresses that occur, this steel is used exclusively in the quenched and tempered condition. Required properties such as high high-temperature strength and high hardness combined with high toughness are, in principle, contradictory and can only be adjusted consistently over the entire die by furnace-based heat treatment. However, the results of various investigations have shown that improvements in the thermal shock resistance and wear resistance of hot work tool steels can be achieved by thermally influencing the microstructure near the surface. Based on these studies and related findings, an approach to surface heat treatment using the electron beam was developed. Due to the particle character of the radiation and the associated possibility of high-frequency beam deflection, the electron beam offers significantly greater flexibility in energy input into the workpiece surface compared with lasers or induction. The overall technological concept envisages replacing furnace-based heat treatment in the production of casting dies by localized and demand-oriented boundary layer heat treatment with the electron beam. The experiments include, on the one hand, the experimental determination of a suitable temperature–time interval with a focus on short-term austenitization. On the other hand, a simulation-based approach of boundary layer heat treatment with validation of a suitable heat source is investigated. Regarding short-term austenitization, the corresponding temperature and time range could be narrowed down more precisely. Some of these parameter combinations seem to be very suitable for practical use. The test specimens show a hard surface layer with a depth of at least up to 6 mm and a very tough buffer layer. Numerical simulation is used to estimate the resulting metallurgical microstructure and the achievable hardness as a function of the temperature–time interval. In addition, the results provided show the possibility of determining and optimizing the material properties by means of a simulation-based approach within the framework of a purely digital process planning and subsequently transferring them into a process planning. In the technical implementation, a temperature control was first established by means of a two-color pyrometer. In the further course of research, the pyrometer will be supplemented by an internally installed infrared camera, which will allow the reproducible setting of specified temperature profiles even for complex, large-area contours in the future.

**Keywords:** die casting; hot work tool steel; simulation; heat treatment; electron beam; near-surface layer

## 1. Introduction

Die casting die are generally subjected to high cyclical thermal and mechanical stresses because of the casting process. The thermal stresses result from the temperature difference between the hot aluminum melt, the release agent to be applied and the mostly cooled die. In addition, mechanical stresses are applied by the opening and closing of the die halves and

the compression of the molten aluminum during the holding pressure phase. Superimposed on the thermally induced and mechanical stresses are chemical and tribological interactions between the melt and the die surface. These processes lead to damage in various areas of the die. In this context, three main damage patterns are described, the so-called stress cracks, thermal cracks and erosion [1,2].

In order to withstand the described stresses, iron-based die materials, such as the hot work tool steel X37CrMoV5-1 (hereafter referred to as H11), are used exclusively in the quenched and tempered condition. A target hardness of $440 \pm 20$ HV has proven to be optimal, which is set by an appropriately adapted heat treatment [1,3]. The state of the art is furnace-based heat treatment, consisting of austenitization, quenching and usually multistage quenching and tempering processes [4]. However, in furnace-based heat treatment, there is often a conflict of objectives with respect to the desired material behavior. The demands for high high-temperature strength and high hardness combined with high toughness are, in principle, contradictory and can only be met as a compromise in the quenched and tempered state of the material. Higher tempering temperatures, in conjunction with suitable austenitizing temperatures, lead to higher fracture toughness values with simultaneously lower hardness values of the hot work tool steel [5,6]. Regarding the achievable service life, however, investigations have shown that high hardness values of approx. 600 HV lead to a longer service life when H11 is subjected to dynamic loading. However, it is evident from the literature that cracking occurs after only 1% of the total service life [7]. Therefore, the requirement for the tool material to have the highest possible toughness properties cannot be neglected. Tribologically induced interactions, for their part, require the highest possible hardness at the die surface [8,9].

## 2. Literature Review

The results of various research projects show that improvements in the thermal shock resistance and wear resistance of hot work tool steels of steel grade H11 can be achieved by selective thermal or chemical influencing of the near-surface microstructure. Approaches here are thermally induced surface hardening as a solid-phase process, remelt alloying as a liquid-phase process or nitriding as a thermochemical process. The solid-phase processes are generally based on short heating or holding times and rapid self-cooling in the material in order to induce martensitic transformations of the near-surface areas. In the liquid-phase process, on the other hand, the starting material is melted and, at the same time, a filler material is introduced into the molten bath. Lasers are primarily used as the energy source for the solid- and liquid-phase processes [10–14]. The group around Telasang et al. sees the radiation energy density $(J/mm^2)$ as a decisive criterion for influencing the metallurgical/mechanical properties of the material. Depending on different radiation energy densities, material samples were taken from heat-treated areas for micro tensile tests. For a radiation energy density of 62.5 $J/mm^2$, a high tensile strength of 2290 MPa could be achieved with a simultaneous elongation at break of 3.6% [10].

In the solid-state process, the energy can also be introduced by induction. However, for various reasons, the knowledge gained has found little application in industrial die making. First, there is the problem that steep flanks of the die casting die can only be heat-treated to a very limited extent due to the properties of the laser radiation. The reflections occurring here would have to be suppressed by an additional coating of the surface. In addition, there is the challenge that during remelting in these sloped areas, the resulting beads are difficult to control due to the melt pool dynamics. On the other hand, a complex die contour requires complex control or tracking of the laser beam [15–17]. A similar problem occurs with induction surface heat treatment. A specially shaped inductor would have to be manufactured for each die geometry, which significantly limits the flexibility of this process. In addition, there is no inductor geometry that can be used to uniformly heat a complexly contoured die casting die. Nitriding does have advantages in terms of wear resistance and adhesion tendency, but the tendency for fire cracking increases. Due to

the high hardness and low toughness of the surface layer compared to the base material, nitrided surface layers are more prone to failure [18].

The depth of the surface-layer heat treatment is of great importance for the achievable service life of the casting tool. Information on the depth of applied boundary layer modification can be obtained from various concepts and technical approaches. For example, [19] reports a depth of hardening (EHT) of 0.15 mm in his studies on gas nitriding of hot work steel H13. The scientists around Cong et al. achieved an EHT of approximately 400 µm during local melting or alloying of hot work steel H11 [20]. In [21], graded material properties were produced for the base material H13 by laser deposition welding. The thickness of these deposition-welded areas was approximately 1 mm. In [22], the authors performed an inventory of near-surface defects using a die-cast housing component. For this component, mean crack depths of 0.5 mm to 2 mm were experimentally determined after 35,000 casting cycles at 12 different positions. As part of his investigations, Müller also looked in detail at the die stresses occurring during filling, solidification and spraying. It was shown that the resulting compressive and tensile stresses are halved already after a depth of 2 mm to 4 mm [2].

Petrov and his team are intensively studying the possibility of surface modification by electron beam for steels and other metallic materials. In doing so, they have developed a numerical model based on the solution of the heat transfer equation using Green functions and used it to calculate the temperature field caused by high-frequency electron-beam scanning [23–25].

Buchwalder et al. developed the WIAS-SHarP software for simulating the surface hardening of steel with laser and electron beams as part of a two-year interdisciplinary research project. They described a mathematical model for the surface hardening of steel consisting of a system of ordinary differential equations describing the microstructural transformations coupled with a quasilinear heat conduction equation. Again, the steel H11 was investigated [26].

In the mentioned boundary layer modifications, short-term austenitization prior to rapid internal cooling of the material plays a crucial role. Austenitization is generally based on diffusion processes in the material, which are significantly influenced by time and temperature. In short-term austenitization, the achievable austenitization temperature is the driving force and is raised by approximately 50 K to 100 K by the characteristically fast heating rates during surface heat treatment [27]. In addition, the initial microstructure strongly influences the austenitizing conditions [28]. A soft-annealed coarse-grained microstructure requires significantly higher temperatures for complete austenitization than a homogenized or quenched and tempered microstructure. This can be explained by the inhomogeneously distributed carbon content and the associated longer diffusion paths of the carbon. In the case of H11 steel, the special carbides involved in short-term austenitization must also be considered. These carbides are deliberately introduced to ensure the high-temperature strength of the steels. However, they complicate the austenitizing behavior, especially in the lower temperature ranges, since grain growth is inhibited during austenitizing. Appropriate adjustment of temperatures must be considered [27].

## 3. Concept

With regard to the experience and findings on the lifetime of die casting dies and the results obtained by other scientists on H11 surface modifications, approaches to surface heat treatment are being investigated which can be achieved by means of electron beams (EB) [29,30].

The overall technological concept is thus aimed at replacing the conventional furnace-based heat treatment in the production of die casting dies by a local and demand-oriented surface heat treatment using electron beams. In accordance with the requirements profile, areas with a high risk of damage (compare Figure 1a) and depth profiles of the die have to be modified to resist the occurring damage mechanisms and thus prevent the typical damages (compare Figure 1b). However, this requires the presence of an already pre-

hardened material in order to ensure sufficient basic strength of the die insert. This can be realized with pre-hardened semi-finished products that are already manufactured to final dimensions before surface heat treatment. Thus, subsequent machining or cost-intensive spark erosion to the final shape of the cavity, which would otherwise be necessary, is no longer required.

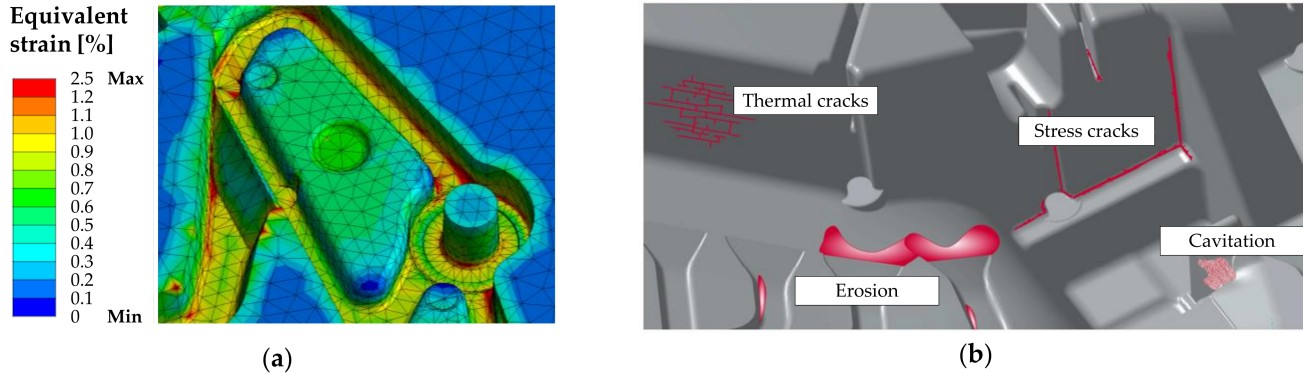

(**a**)                                                              (**b**)

**Figure 1.** A simulated result of a lifetime calculation to determine critical die contours (**a**) and schematic representation of a die contour with the main described damages and their typical related occurring areas (**b**).

Figure 2 shows the process flow of a conventional as well as a novel electron-beam-based heat treatment. It becomes clear that the manufacturing steps required to produce a final contoured, heat-treated die insert are reduced compared to a conventional heat treatment. Nevertheless, as a result of the new degrees of freedom regarding temperature–time profiles, additional data or decision-making bases must exist for a local heat treatment to be carried out in a targeted manner. This leads to the concept of an extended, digital process planning for the casting tool, which is also shown in Figure 2.

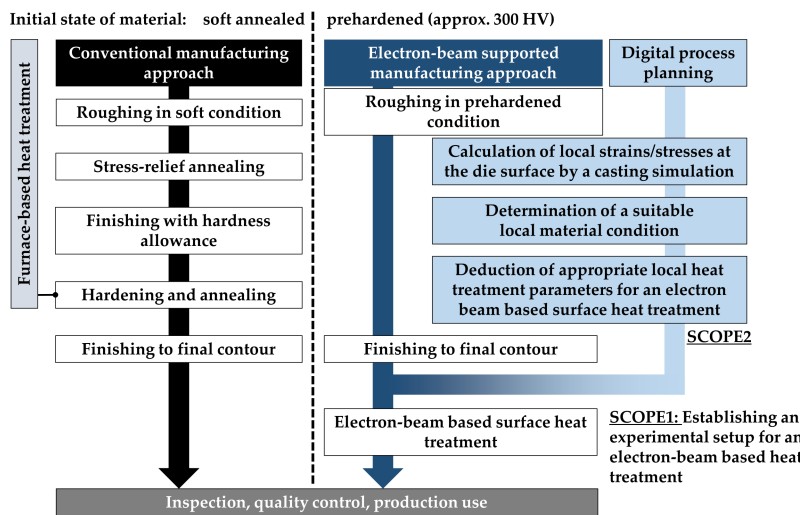

**Figure 2.** Manufacturing concept of an electron-beam supported heat treatment compared to a conventional, furnace-based heat treatment; designation of Scope 1 and 2 considered within the framework of this article.

In accordance with the requirements definition, areas at risk of damage must be identified first. This is done by means of criteria derived from casting simulations, e.g., local strains [2] or flow velocities, and will not be discussed in further detail here.

Based on the derived criteria, suitable material properties are to be determined which, with respect to their property profile, anticipate the causes of failure or reduce the damage intensity as far as possible.

Based on the required properties, suitable heat treatment parameters—i.e., locally different time-temperature profiles at the surface—are to be defined. Since both deeper areas and complex geometries must be heat-treated and the increasing heating of the tool also has to be taken into account, this can only be done using a simulation-based approach.

As a result, the heat treatment of the die can be designed and carried out as required within the framework of a digital process planning; the absence of a furnace-based heat treatment eliminates further manufacturing steps that are associated with risks.

Figure 2 also lists the two core scopes (Scope 1 and 2) considered in this article. Scope 1 contains the presentation of the experimental realization of the concept. In Scope 2, the simulation-based approach for characteristic value determination and process planning is explained in more detail. In this context, Figure 3 shows the logical relationships among each other for Scope 2 and the assignment to the respective chapters within the framework of this publication.

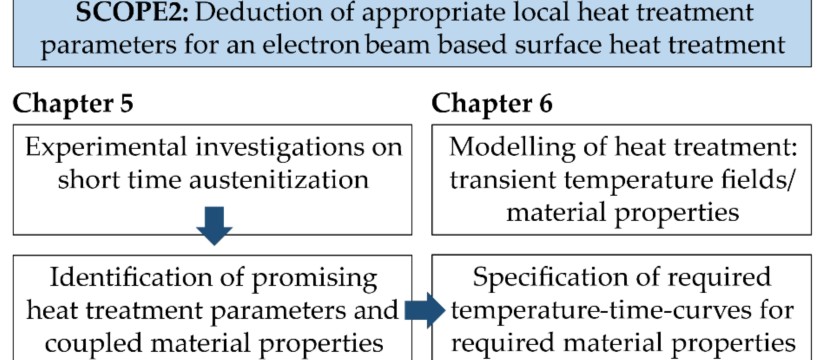

**Figure 3.** Combining experimental parameter determination and coupled thermal simulation/material simulation to realize a digital, simulation-based process planning.

### 4. Experimental Setup

Within the scope of the research, all experimental tests were performed with an electron-beam system of the type K26-3, manufactured by pro-beam AG & Co. KGaA (Düsseldorf, Germany). The system is designed according to the chamber principle. This means that the available working space consists of a recipient, which must be opened completely when changing workpieces or tools. The chamber volume of the recipient is 2.6 m$^3$, which provides sufficient space to process even larger components. With the aid of the pump system, the vacuum chamber can be evacuated in continuous operation in approx. 420 s to a vacuum of $10^{-6}$ bar required for the process. The pieces are positioned on a worktable that can be moved in the X-Y plane by a computer numerical control (CNC) system. In addition, a pyrometer is installed inside the working chamber. The heating process can be monitored with an installed optical charge-coupled device camera (CCD camera) on the one hand and with the so-called electron optical image (ELO image) on the other. In addition to the working chamber, the entire system includes the electron-beam generator, which can generate a maximum accelerating voltage of 150 kV for the present system. The resulting maximum beam power is 15 kW. In order to prevent heat accumulation in the clamping, additional copper plates with water flowing through them were integrated into the chamber.

Preliminary tests have shown that the possibilities for variation are very extensive when finding suitable system parameters [31]. In addition, working with constant parameters, such as the beam current, does not always make sense. In some cases, this resulted in melting of the sample surface. It is therefore advisable to break down the heating process to the parameters of temperature and holding time or feed rate. In the further technical implementation, a temperature control was thus established by means of a 2-color pyrometer, analogous to the schematic representation in Figure 4a. Due to the 2-channel technique, the 2-color pyrometer can, in principle, be operated independently of the emission coeffi-

cient and is therefore very well-suited for heating processes of bright steel surfaces. The pyrometer can control the current set beam power $I_B$ via a proportional–integral–derivative controller (PID controller). The controller converts the temperature into a voltage signal and passes this to the electron-beam machine as a manipulated variable between 0 V and 10 V. A scale of the maximum set beam current $I_B$ to the range of the manipulated variable is required before heating. The entire real setup of the electron-beam machine with the pyrometer can be seen in Figure 4b. The thermographic camera with the housing shown is used for further experiments based on the current findings.

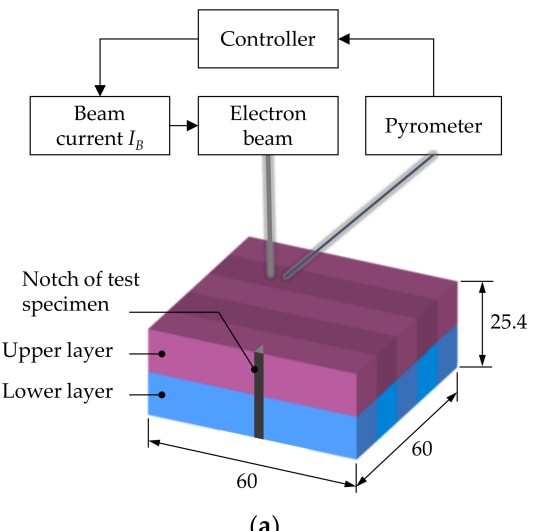

(a)

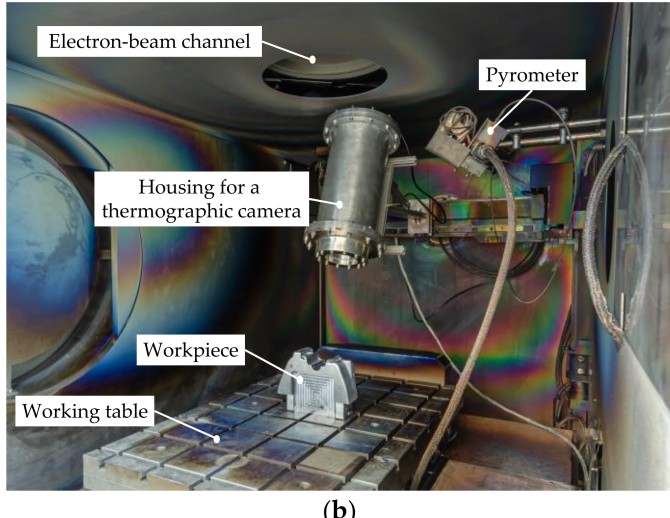

(b)

**Figure 4.** Schematic simple block diagram of the closed-loop control (**a**) and the real setup of the electron-beam machine (**b**).

## 5. Experimental Investigations

The aims of the following investigation are, on the one hand, to better understand the effect of short-term austenitization of pre-hardened samples from the H11 (compare chemical composition in Table 1) and, on the other hand, to identify a suitable time–temperature profile to obtain the required mechanical properties. This was done by varying the parameters of holding time and holding temperature during the surface-layer heat treatment. The holding temperature was increased from 1000 °C in three steps to 1200 °C, whereby three holding times of 10 s to 60 s were set. The samples had a cuboid shape with dimensions of 60 mm × 60 mm × 25.4 mm. Each sample was pre-hardened to a hardness of 330 HV125 according to the specifications (furnace with nitrogen fumigation, austenitization: 1000 °C/30 min, quenching with compressed air, annealing: 650 °C/120 min with air cooling (one time)) of the material supplier. The temperatures were recorded during the heating phase and the cooling phase due to the pyrometer measuring range from and up to 350 °C respectively. In order to compensate for the temperature jump at the beginning to 350 °C, a short holding step of 5 s at 450 °C was planned (compare Figure 4). The heating rate was 10 K/s. In all tests, only solid-phase processes were considered, whereby a relative movement between the workpiece and the electron beam did not take place.

**Table 1.** Default and measured chemical composition of the H11 test material, measured with the optical emission spectroscopy device SPECTROLAB by the SPECTRO Analytical Instruments GmbH.

| H11 | Chemical Composition (wt.%) | | | | | | | | | |
|---|---|---|---|---|---|---|---|---|---|---|
| | C | Cr | Mo | Ni | Si | Mn | V | S | P | Fe |
| Default | 0.33–0.41 | 4.80–5.50 | 1.10–1.50 | <0.40 | 0.80–1.20 | 0.25–0.50 | 0.30–0.50 | <0.005 | <0.02 | Rest |
| Measured | 0.357 | 4.837 | 1.192 | 0.293 | 1.088 | 0.391 | 0.437 | 0.001 | 0.013 | Rest |

The following EB and controller parameters were set for the experiments:

- Controller:

    – Proportional band $X_P = 0.1\%$
    – Integral time $t_i = 10$ ms
    – Derivative time $t_d = 0$ ms

- EB parameters:

    – Acceleration voltage SH = 120 kV
    – Maximum beam current IB, max = 60 mA
    – Deflection figure squared scanning field
    – Deflection amplitudes X = Y = 60 mm
    – Deflection frequency fB = 1000 Hz

The EB parameters as well as the controller parameters were analyzed and determined in extensive preliminary research. The proportional band as a control parameter, for example, must be selected in such a way that the controlled variable is sufficiently amplified to reach the set temperature at all. Integral time and derivative time prevent excessive overshooting of the controlled variable and thus possible melting of the surface.

The max. beam current and the acceleration voltage were selected in such a way that sufficient energy was available in the control loop for heating up to 1200 °C. The deflection amplitude corresponds to the surface of the components to be treated. Figure 5 shows an example of the measured temperature curve of the parameter variation 10 s holding time and 1000 °C holding temperature. A slight alternation of the actual temperature around the set temperature can be clearly seen in the detailed image in Figure 5. The variation is in a range of approximately ±5 K and thus tolerable. This is caused by the control loop and the highly transient energy input by the electron beam. Despite extensive adjustment of the control parameters, the fluctuation range could not be reduced. A faster PID controller would be required in this case.

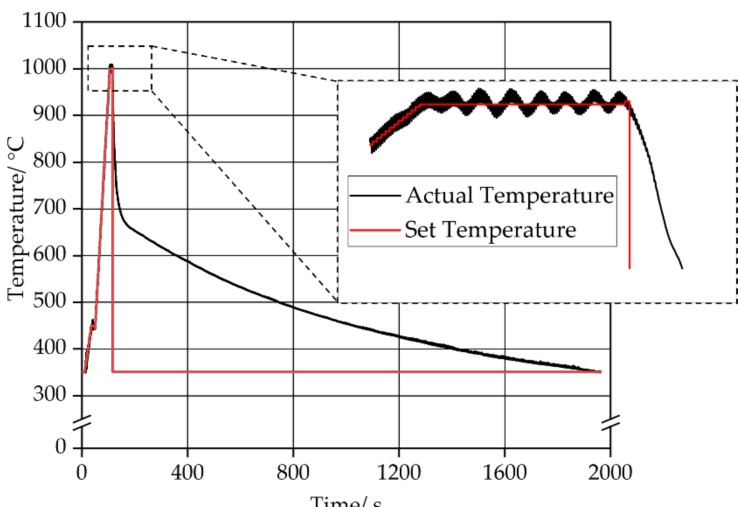

**Figure 5.** Example of a measured temperature curve, holding for 10 s at 1000 °C.

For better comparison, the $T_{85}$-time was determined with the recorded temperature curves. Although this is mainly used for welding processes with the associated continuous cooling transformation (CCT) diagrams, it can also be used for heat treatment processes for a better estimation of the resulting hardness. The T85-time is the duration it takes for a body to naturally cool from 800 °C to 500 °C. This parameter is calculated when planning welding operations in order to be able to predict the resulting microstructure when a CCT-diagram is presented [32]. The derived $T_{85}$-times are shown as a red line profile in the diagram in Figure 6a. According to the cooling curves from the CCT diagrams

for the H11 according to [33,34], hardness of more than 600 HV can be expected for all parameter variations. This can be partially confirmed according to the comparison of the obtained hardness from the diagram in Figure 6a. The lower values determined at 60 s holding time and 1100 °C or 1200 °C can be attributed (cf. Figure 6b) to a strong coarse grain in the area close to the surface.

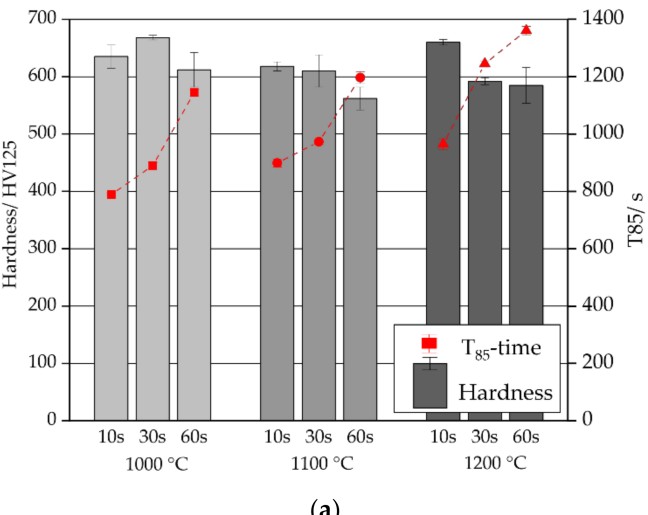

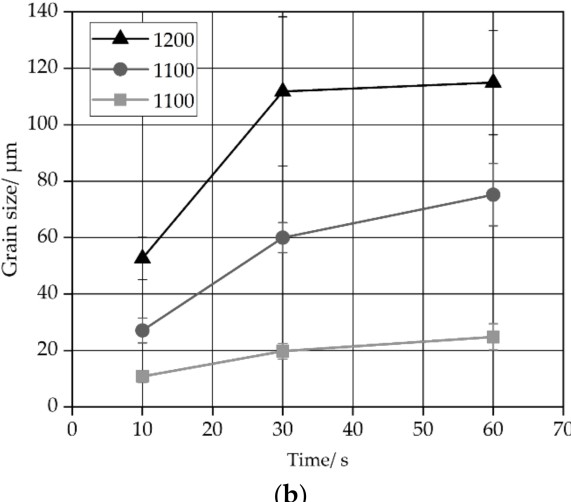

(**a**)                                                                                        (**b**)

**Figure 6.** Comparison of the measured hardness of the surface and the corresponding $T_{85}$-time (**a**) and comparison of the measured average grain size according to the DIN EN ISO 643 standard in the upper layer (**b**).

Furthermore, the variation range of the $T_{85}$-times seems to be too small to show significant dependencies on the resulting hardness. Nevertheless, a significant increase in the hardness compared to the initial hardness of 330 HV125 was achieved regardless of the parameter selection.

According to Figure 6b, it can be noticed that the undesirable coarse grain formation sets in comparatively quickly. If 200 Kelvin higher austenitization temperatures are selected, the average grain diameter quadruples. With a standardized heat treatment at a temperature of 1000 °C, a mean grain diameter of 10 μm to 15 μm can be expected. Compared to the continuous grain growth diagram from the H11 according to [33], the formation of coarse grains seems to shift towards lower temperatures during short-term austenitization [33]. This means that with the set heating rate of 10 K/s and the investigated temperatures between 1000 °C and 1200 °C, smaller grain diameters are normally expected [33]. A temperature increase of more than 100 K should not be selected in order to prevent unwanted coarse grain formation. The grain diameter was determined by the software ImageJ according to the DIN EN ISO 643 standard with the line segment method [35].

In order to estimate the hardening depth, additional random hardness measurements were carried out on cross-sections using the ultrasonic contact impedance (UCI) method with a discrete distance of 0.15 mm between each measurement point. As an example, the false-color images of the hardness profiles for the parameter variations 1000 °C/10 s and 1100 °C/30 s are shown in Figure 7a,b, respectively. As expected, a deeper hardening depth can be seen with increasing temperature and holding time. Hardening depths of 4 mm to 6 mm could be achieved even with relatively short holding times and low austenitization temperatures. Higher temperatures and holding times very quickly lead to heat accumulation, which in turn affects areas of the component that are too deep. It should be mentioned that the depth of the hardness is very likely to be lower for components with larger volumes compared to the tests carried out here.

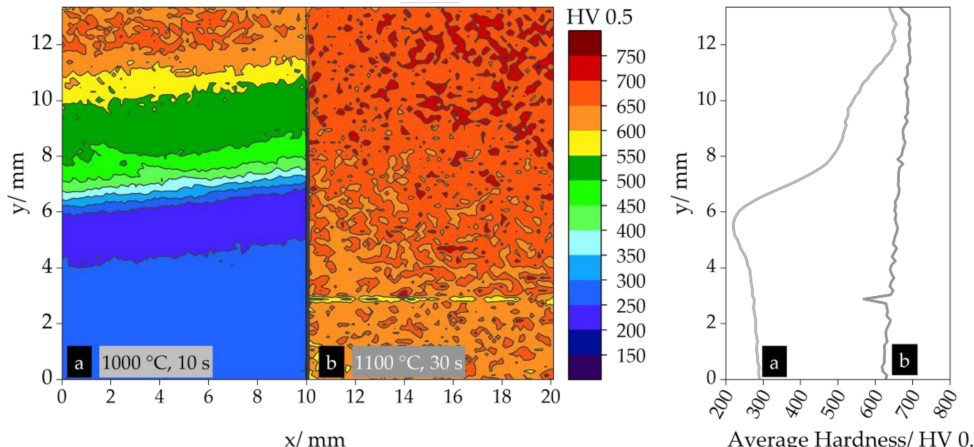

**Figure 7.** Two false-color images of a planar hardness profile 1000 °C, 10 s (**a**) and 1100 °C, 30 s (**b**).

Figure 8 shows one of the cross-sections of the near-surface layer of the sample 1000 °C/10 s. The cross-sections were etched with an alcoholic nitric acid and analyzed with an optical microscope. It shows a fine-grained martensitic area with a local hardness increase (cf. red/orange-colored at 12 mm on y-axis in Figure 5) and some annealing effects (annealed martensite with many carbides) in the lower region (cf. light-blue-colored at 2 mm on y-axis in Figure 5).

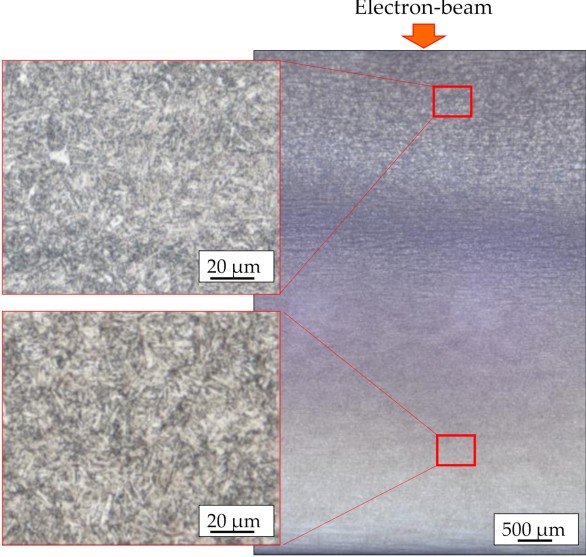

**Figure 8.** Cross-section polishes of the sample heat-treated with the parameters 1000 °C and 10 s.

In addition to the average grain size, the retained austenite content was also measured for all samples in the affected area. As expected from the relatively low carbon content of 0.37% by weight, this was a maximum of 2.5% of retained austenite. An influence on the results of the hardness measurement and the determination of the notched bar impact energy by excessive retained austenite content can therefore be excluded.

Finally, the notched bar impact work was determined for the samples. Notched bar impact test specimens were taken from both the upper and lower areas. The test was carried out at room temperature in accordance with the DIN EN ISO 148 standards [36]. The results are shown in the diagrams in Figure 9a,b. The sample positions are shown in Figure 4a.

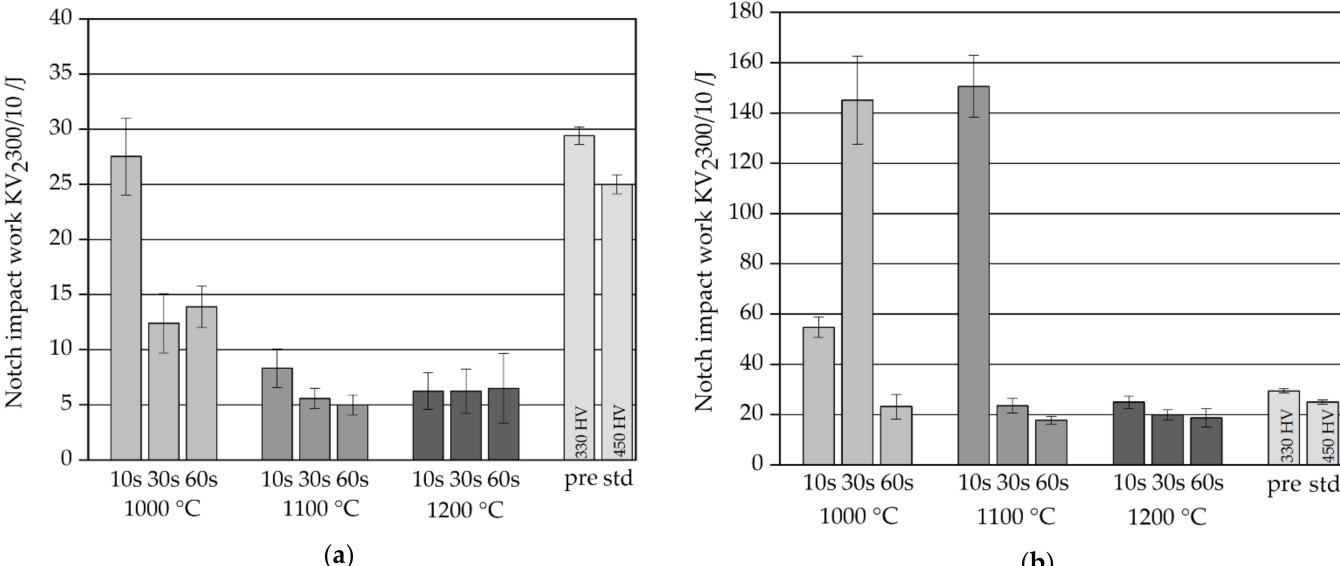

**Figure 9.** Measured values of the notch impact work at room temperature according to the DIN EN ISO 148 standard in the upper layer (**a**) and in the lower layer (**b**).

For comparison, the averaged value from the pre-hardened material (designated "pre") and a standard heat treatment up to 450 HV (designated "std") were included in the diagrams. The values from the near-surface samples are very low regardless of the treatment parameters. The exception is the combination 1000 °C/10 s. Here, a value similar to that expected from a standard annealing to approx. 440 HV (25.2 J) is shown [31]. All values are below the pre-hardened material. A similar behavior is shown by the samples from the lower layers. The exception is the combination 1000 °C/30 s and 1100 °C/10 s. Here, the material seems to have been tempered over a relatively larger volume by the corresponding temperature ranges during the surface heat treatment. The notched bar impact work obtained clearly exceeds the initial value or that of the standard tempering. In combination with the values from the upper layer, these two parameter combinations seem to be sufficient to obtain the required microstructure and the corresponding mechanical properties. The test pieces show a hard surface layer with a depth of at least 6 mm and a very tough buffer layer.

Regarding the achievable lifetime of die casting dies, a tough material behavior has a decisive influence. Based on Hihara's investigations, residual stresses caused by thermal cycling occur after comparatively few cycles. By superposition of residual stress and process-related stresses, the yield strength of the material is locally exceeded, and the residual stress is reduced by deformation. The result is the formation of cracks in the die surface. An appropriately adjusted toughness in the material can greatly minimize the crack propagation over the required number of casting cycles and thus contribute to increasing the lifetime of the die [37].

## 6. Simulation-Based Approach

Although suitable EB parameters for a target hardness or notch impact energy at the surface can be easily determined with the help of experimental tests, they provide a statement about the temperature distribution inside the material as well as the accompanying material processes only insufficiently or with a high testing effort. Since hardness and impact energy are time- and temperature-dependent during heat treatment processes, a simulative estimate can be used by determining the corresponding temperature distribution. In general, the aim is to identify a suitable temperature–time profile for both the surface and the internal material areas. For this purpose, a simulative observation of the temperature distribution is suitable, which in turn is modeled by transient temperature fields. These temperature fields are used to estimate the resulting metallurgical structure

and the hardness profile that can be achieved. This is done by the finite element method (FEM) open-source software Code_Aster (version 15.3, Paris, France). As an open system, Code_Aster offers extensive variations for the creation of different substitute heat sources. The code is based on Python programming language. If the calculation cases are implemented, e.g., script-based and not via the associated GUI Salome-Meca, in principle, any mathematical function can be used which can be calculated in the Python module Numpy. In addition, the implementation of the metallurgical effects according to the model of Leblond and Devaux integrates a very mature method. Another advantage is the extensive community, which can quickly contribute to a solution when problems arise.

To simplify the elaborated model, the movement of the single focused electron beam is not considered as such. Instead, the resulting area of the deflected beam is assumed as a coherent heat source. This is justified by the highly transient processes in the beam deflection (which can be up to 1 MHz) and the associated very short time steps in the simulation. Superimposed on the required fine meshing of the treated areas, this would result in disproportionately long computing times.

The description of the heat distribution is based on Fourier's law of heat conduction with the partial differential equation [38]:

$$\rho c(T)\frac{\partial T}{\partial t} - \lambda(T)\Delta T + \frac{d\lambda}{dT}grad^2 T = \dot{W} \quad with\ \rho = const.\ and\ \dot{W} \equiv 0 \tag{1}$$

For the specific heat capacity and the thermal conductivity, a temperature-dependent behavior is considered. The values for H11 were taken from [2]. The calculations are therefore non-linear. Regarding these material properties, a homogeneous state is assumed at time $t_0$. The heat input can be considered in Code_Aster via a second-type boundary condition, among others, with [38]:

$$\dot{q} = -\lambda\frac{\partial T}{\partial n} \tag{2}$$

Initially, this is assumed to be independent of temperature and location. When the thermal imaging camera is put into operation, the extent to which the heat flow is dependent on location is to be checked. In this case, the model would be adapted. The required values are derived from the radiant power and the incident surface. The heat flux density again acts normal to the impinging surface. The shape corresponds to the deflection figure resulting from the deflection and is combined as a single group of nodes during meshing. Latent heat during a phase transition is not considered, since only solid-phase processes are involved. As a further boundary condition, a constant temperature is assumed at the bottom of the block, which in turn is intended to simulate the active cooling of the block by the heat sinks. Convective heat losses or thermal radiation are not considered.

To enable a transient calculation, a corresponding time discrepancy must be made in Code_Aster by specifying a time sequence with time steps. Furthermore, at each starting time of the time step to be calculated, the calculated temperature field from the previous step is used as an initial condition. The resulting structure is thus enriched with each further calculated time step. This is exactly what makes the simulation of a moving heat source possible. This is achieved by modulating a certain number of groups of nodes according to the feed rate, on which the heat flux density acts for a time limit. The meshing of the models is done by means of the algorithm NETGEN 1D-2D-3D from GMESH, using tetrahedral elements of the type TETRA4 with a strong refinement of the mesh at the respective surface.

For the calculation of the microstructure, the models according to Leblond and Devaux are used for diffusion-controlled transformations of the microstructure during heating and cooling [39]:

$$\frac{dP_i}{dt} = n_{ij} \left( \frac{K_{ij}P_i - K'_{ij}P_j}{Tr} \right) \left( ln \left( \frac{K_{ij}(P_i + P_j)}{K_{ij}P_i - K'_{ij}P_j} \right) \right)^{\frac{n-1}{n}} \tag{3}$$

The model is based on the well-known Johnson–Mehl–Avrami equation (JMAG), which can be used for diffusion-controlled isothermal phase transformation. In order to take non-isothermal processes into account, a correction value is introduced in the equation of Leblond and Devaux, which takes the influence of the cooling rate into account. It can be used for the transformation processes during heating as well as for the processes during cooling after austenitization. In addition, the influence of grain growth on the transformation kinetics is considered in the simulation by the Arrhenius equation [39]:

$$\frac{dD^a}{dt} = Ce^{(-\frac{Q}{RT})} \tag{4}$$

The diffusionless folding process during martensite formation is calculated according to the model of Koistinen and Marbuger, corresponding to the following equation [39]:

$$p_i(T) = p_{i,eq}(1 - e^{(-b(M_s - T))}) \tag{5}$$

The resulting hardness can finally be interpolated via the calculated phase fractions of ferrite, bainite, pearlite and martensite as well as their individual hardness values. The cooling curves with the resulting microstructure fractions as well as the values for the X37CrMoV5-1 with the respective expected microstructure fractions are taken from [33,34] and implemented with a Fortran routine in Code_Aster.

For the validation of the model, the temperature fields during the boundary-layer heat treatment are measured and compared with the results of the simulation (compare Figure 10a,b). Step by step, the substitute heat sources are calibrated in the simulation. In addition, a comparison is made via metallographic cross-sections and measurements of the hardness distribution (cf. Figure 10c,d). The validation of the model is initially carried out on flat test samples with a size of 40 mm × 40 mm × 25.4 mm (compare subfigure in Figure 10a).

Figure 11a shows, finally, a comparison of a calculated hardness distribution in the cross-section after a statically performed heat treatment after cooling down to room temperature. The sample (cf. Figure 4a) was heated up with a similar rate to the described experiment up to 1000 °C and held for 10 s and 30 s. The corresponding simulated cooling curves are shown in Figure 11b. The significant cooling rates between 800 °C and 500 °C can be derived from the simulated cooling curves. The combination 1000 °C/10 s results in a rate of 0.36 K/s and the 1000 °C/30 s in a rate of 0.27 K/s.

Currently, the model needs to be further adapted and improved. Further experimental and simulated results will show whether the model used to calculate the microstructure is sufficiently accurate in depicting the material science relationships, even when considering the prevailing cooling curves in the case of short-term austenitization, as is the case with surface hardening. In the further course of the project, the results will be transferred to more complex surfaces and die inserts.

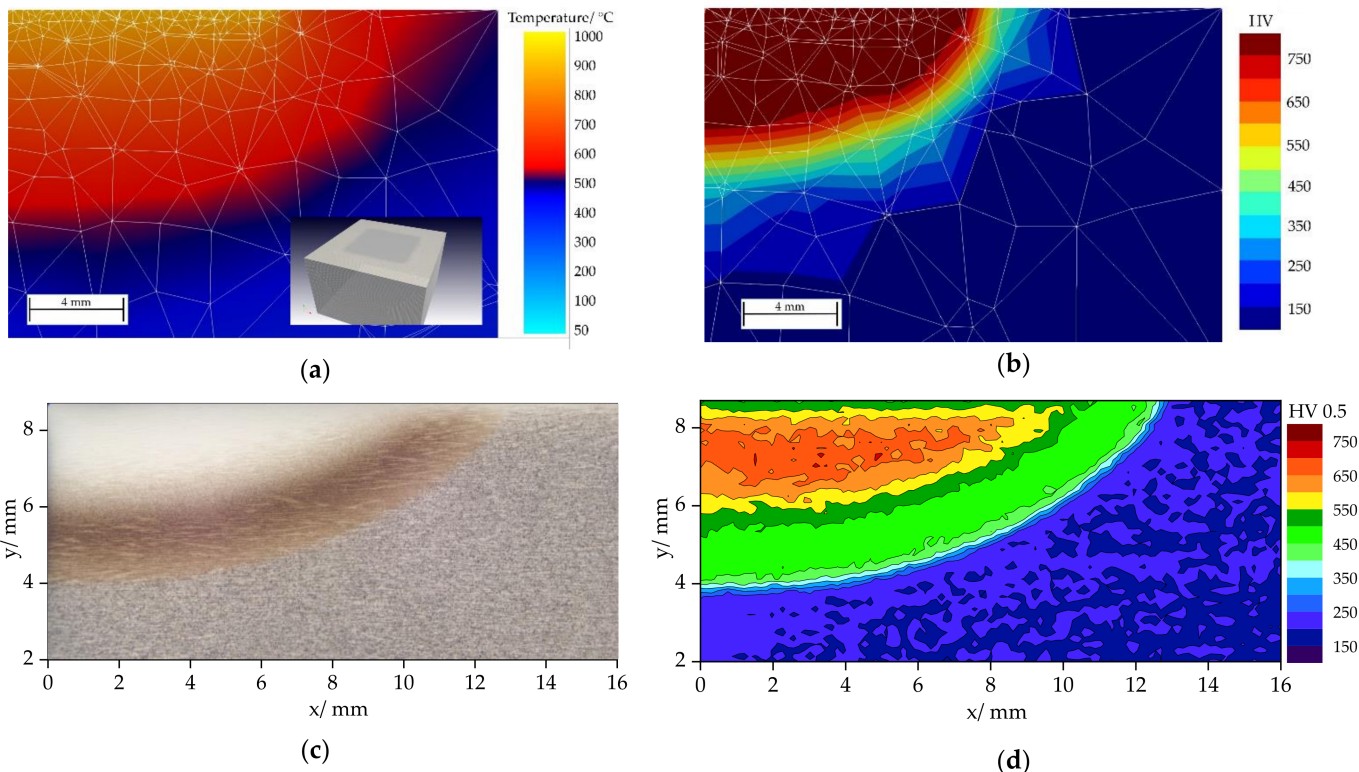

**Figure 10.** Example of a simulated temperature distribution (sample size 40 mm × 40 mm × 25.4 mm) during heating using static transient heat source (**a**), the belonging calculated hardness distribution after cooling down (**b**), the corresponding cross-section polish of the real treated sample (**c**) and the belonging two-dimensional hardness distribution measured with UCI (**d**).

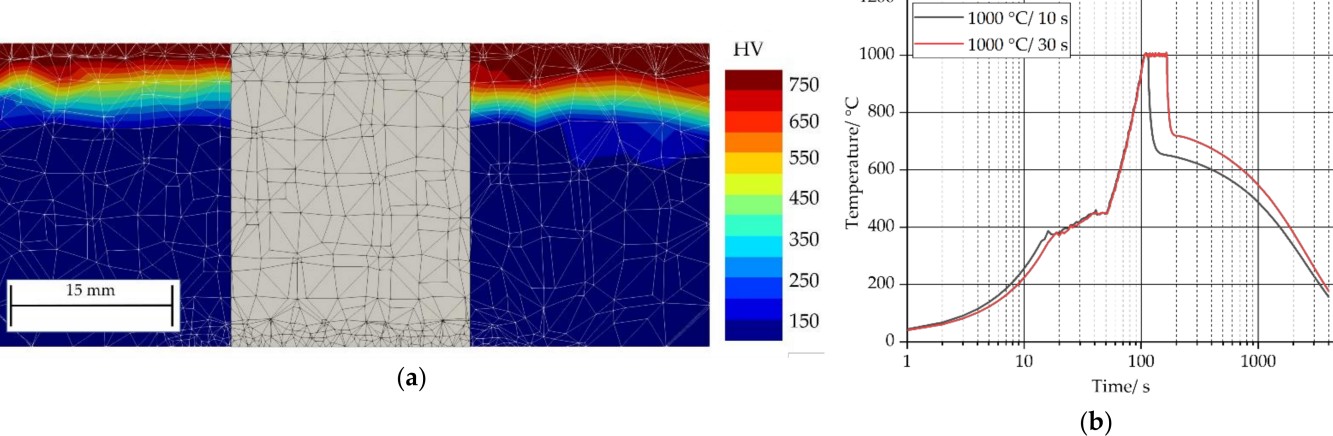

**Figure 11.** Comparison of a calculated hardness distribution within a cross-section after cooling down, left parameter combination 1000 °C, 10 s and right 1000 °C, 30 s (middle shows the mesh size) (**a**) and the corresponding simulated cooling curves (**b**).

## 7. Summary and Further Procedure

Practical tests have shown that hardening depths of up to 12 mm are possible by means of suitable control, depending on the specimen geometry. Hardness values of up to 650 HV in the near-surface areas could be achieved on pre-hardened samples with an initial hardness of approx. 330 HV. Regarding the short-term austenitization, the corresponding temperature and time range could be narrowed down more precisely. The resulting microstructure is finely martensitic in the treated areas near the surface and thus also very brittle. In this case, a subsequent tempering process is also necessary

regarding the precipitation of the important secondary carbides. The extent to which such tempering processes can be realized with the electron beam was also the subject of current investigations. In summary, it can be said that local heat treatment can produce cooling rates that are either technically not possible in conventional, furnace-based heat treatment or are not feasible due to hardening cracks or distortion due to excessive manufacturing risk. Based on the modified temperature control, it is assumed that a service life increases of approximately 20% can be achieved compared to conventional furnace-based heat treatment. This assumption, which refers to thermally induced defects on the die surface, is justified by the fact that higher hardness on the die surface and thus higher tolerable stress ranges can be achieved while maintaining the same ductility as far as possible. In combination with the simulation-supported models for predicting the known damage mechanisms, a valuable contribution can thus be made to increasing the economic efficiency of the die casting process.

In the further course of the work, the pyrometer will be supplemented by an internally installed thermal imaging camera, which will, in future, allow temperature profiles to be set even for complex, large-area contours. The thermal imaging camera will also enable a more precise determination of the substitute heat source during the simulation. Furthermore, a control loop is being worked on, which is to be realized with the infrared camera. This should make it possible to heat-treat contoured surfaces according to given specifications.

**Author Contributions:** Conceptualization, T.S. and S.M.; methodology, T.S. and S.M.; software, T.S.; validation, T.S. and S.M.; formal analysis, T.S.; investigation, T.S.; resources, T.S.; data curation, T.S.; writing—original draft preparation, T.S.; writing—review and editing, S.M. and K.D.; visualization, T.S. and S.M.; supervision, K.D.; project administration, S.M.; funding acquisition, T.S. and S.M. All authors have read and agreed to the published version of the manuscript.

**Funding:** The IG research project 20552N of the Research Association Foundry Technology FVG is funded by the Federal Ministry of Economics and Technology via the AiF within the framework of the programme for the promotion of joint industrial research and development.

**Institutional Review Board Statement:** Not applicable.

**Informed Consent Statement:** Not applicable.

**Data Availability Statement:** Not applicable.

**Acknowledgments:** The authors would like to thank the members of the IG research project 20552N committee for their support and cooperation.

**Conflicts of Interest:** The authors declare no conflict of interest.

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
