# Peer review of "A Near-Surface Layer Heat Treatment of Die Casting Dies by Means of Electron-Beam Technology"

_metals, doi:10.3390/met11081236_

Round 1

Reviewer 1 Report

In the manuscript entitled “A simulation-supported surface-layer heat treatment of die casting dies by means of electron-beam technology” by T. Schuchardt et al, the authors investigate the approach to surface heat treatment by means of electron beam to improve mechanism properties and subsequently increase the lifetime of die casting dies. It is significant and might attract interest from the readers of Metals. However, before publishing this work, it is recommended to address follow comments.

1. The abbreviations and symbols should be expanded or explicated at first use, such as EDM, CNC control system, PID-controller, T85-time. What’s T85-time and what does T85-time mean?

2. There are two lines in Figure 4, which one represents set temperature.

3. What the reason for low surface hardness at 30 s holding time and 1200 ºC.

4. Figure 6(a) should be replaced by an image with higher quality.

5. What does “the formation of coarse grains seems to shift towards lower temperatures during short-term austenitisation (cf. [32, 33])” mean.

6. The authors claimed “A temperature increase of more than 100K should not be selected in any case.” Why?

7. In Line 274, the authors claimed that “The test pieces show a hard surface layer with a depth of at least to 6 mm and a very tough buffer layer.” It is more convincing if the authors can give the hardness profiles for the combination 1000 ºC/30s and 1100 ºC/10s.

8. In Figure 9 and 10, a scale bar should be added.

9. There are few discussions on the validity and accuracy of the model although the author compared measured temperature and simulated temperature, and compared the measured hardness distribution and calculated hardness distribution.

10. There are some minor grammatical errors in the manuscript including the following:

  • In Figure 2, “EB supportet production” should be “EB supported production”;
  • What is “surface hardness’s”;

Author Response

Thank you for your review and very useful advices you gave us. Some changes we made according yours and the other review. Please find the following changes.

1->we add the expansion and an explanation of theT85-time (page 8)

2->we change the colour of one line for a better identification

3->we suppose that the figure 7b was not very clear (what was the T85 and what the hardness), we changed it, the hardness of 30s and 1200 °C are comparable to the other ones

4->we replaced it

5->we add an explanation, see line 297,  according to the diagram in [32] normally smaller grain-size are expected with the set temperature and rate

6-> we add an explanation, line 301

7-> unfortunatley, we hat not enough time after the review to measured this profiles and add it in the reviced script

8-> we add it, see figure 9a and 10 b

9->we add it, see line 413 till 425

Thank you.

Best regards 

Reviewer 2 Report

In the paper entitled „A simulation-supported surface-layer heat treatment of die casting dies by means of electron-beam technology” Authors present the results of studies concerning some aspects of the technology design of the die made in X37CrMoV5-1 tool steel. The reviewed paper is characterized by a good scientific level and the whole presentation is clear. Moreover, the obtained results have the potential for application in industrial conditions. Therefore, the paper needs only minor revision. The detailed comments are given below:

#1: In chapter 5 on page 5, please add the data concerning spectrometer used in researches of steel chemical composition.

#2: In chapter 5 on page 9 the Authors present the results of microstructure examinations of studied steel such as measurements of grain size and phase composition i.e. mentioned the amount of retained austenite. However, in the paper is not present any microstructure of studied tool steel after surface heat treatment. I suggest adding at least an example microstructure of the studied surface layer with a suitable comment in the main text.

Author Response

Thank you for your review and very useful advices you gave us. Some changes we made according yours and the other review. Please find the following changes. 

#1-> we used a optical emission spectroscopy device SPECTROLAB by the SPECTRO Analytical Instruments GmbH

#2->we add a cross-section polish, see figure 6 in revised manussript

Thank you.

Best regard

Reviewer 3 Report

The manuscript needs to be corrected:

  • In the review section "2", it is necessary to consider the literary sources in more detail (see line 76 with an interval of [10-14]);
  • In the section"3" for figure 1b required to decipher all the abbreviations and in the end clearly state the purpose of research (to transfer the purpose from the section "5");
  • In the section with the experimental installation "4" instead of a primitive block diagram of the feedback control to bring photos of the real camera setup with thermometer and add a reference to figure 3;
  • In the experimental section "5", it is necessary to provide photos of experimental samples, specify the equipment for determining the chemical composition of H11 steel, justify the choice of the parameters of the controller and EB, indicate in the photo or diagram the specific transfer mould sites from which samples were made to determine the impact strength of steel (now very abstractly designated - "upper layer" and "lower layer");
  • In the section "6", explain the choice of the Code_Aster software package, be sure to estimate the cooling rates of steel after electron beam treatment and correctly indicate Figures 9a and 9b (instead of 8a and 8b in line 357);
  • In order to emphasize the novelty of the work, it is necessary to specify the steel grade and the phrase "near-surface layer" in the name and keywords, specify specific hardness values in the annotation (see drain 26) and the T/t combination (see drain 29), and add the level of values of the cooling rates of the steel field after electron beam processing and the values of the economic efficiency of the new transfer mould production process in the summary;
  • In conclusion, I would like to recommend a more understandable title of the manuscript: electron-beam heat treatment of the transfer mould near-surface layer made of H11 tool steel.

Author Response

Thank you for your review and very useful advices you gave us. Some changes we made according yours and the other review. Please find the following changes.

#1-> we add it, see line 80 till 85

#2->we changed the whole figure, because it was not clearly

#3->we changed the figure as well and add another one

#4->the sample were simple blocks, we think that there were now further information by adding pictures of the sample, we add the equipment for the spectroscopy (see table 1), see figure 3 for the samples (lowe layer, upper layer)

#5-> we add, see line 347 till 355

#6-> we add it

according your suggestion for changing the title, we would like to change the title, please see the titel of the manuscript, we want to maintain the die casting, because our researches concerns die casting, expecially with their typical damages, according to our experiences transfer moulding mold are not comparable suffered from damages which are typical for Aluminium die casting

Thank you a lot.

Best regards   

Round 2

Reviewer 1 Report

The authors have answered my comment and I recommend this manuscript for publication in Metals.

Reviewer 3 Report

The manuscript has been
sufficiently improved to warrant publication in Metals.